# BNT162b2 or CoronaVac as the Third Dose against Omicron: Neutralizing Antibody Responses among Transplant Recipients Who Had Received Two Doses of CoronaVac

**DOI:** 10.3390/v15071534

**Published:** 2023-07-12

**Authors:** Çiğdem Erol, Zeynep Ece Kuloğlu, Bircan Kayaaslan, Gülen Esken, Adalet Altunsoy, Tayfun Barlas, Güle Çınar, İmran Hasanoğlu, Ebru Oruç, Said İncir, Alpay Azap, Gülten Korkmaz, Dilara Turan Gökçe, Onur Elvan Kırımker, Ezgi Coşkun Yenigün, Erkan Ölçücüoğlu, Ebru Ayvazoğlu Soy, Süleyman Çetinkünar, Özlem Kurt Azap, Füsun Can, Mehmet Haberal

**Affiliations:** 1Department of Infectious Diseases and Clinical Microbiology, Faculty of Medicine, Başkent University, Ankara 06490, Türkiye; ccatalyurekli@yahoo.com (Ç.E.); ozlem.azap@gmail.com (Ö.K.A.); 2Koç University-İşbank Center for Infectious Diseases (KUISCID), Istanbul 34010, Türkiye; zkuloglu20@ku.edu.tr (Z.E.K.); gesken@ku.edu.tr (G.E.); tbarlas@ku.edu.tr (T.B.); 3Graduate School of Health Sciences (GSHS), Koç University, Istanbul 34450, Türkiye; 4Department of Infectious Diseases and Clinical Microbiology, Ankara City Hospital, Ankara 06800, Türkiye; drbican@gmail.com (B.K.); aadalet@yahoo.com (A.A.); imran.solak@gmail.com (İ.H.); 5Department of Infectious Diseases and Clinical Microbiology, Ankara University, Ankara 06230, Türkiye; gbinjune@gmail.com (G.Ç.); alpay.azap@medicine.ankara.edu.tr (A.A.); 6Department of Infectious Diseases and Clinical Microbiology, Adana City Hospital, Adana 01230, Türkiye; ebru.oruc_@hotmail.com; 7Department of Medical Biochemistry, Koç University School of Medicine, Istanbul 34450, Türkiye; sincir@kuh.ku.edu.tr; 8Department of Hematology, Ankara City Hospital, Ankara 06800, Türkiye; ankarasehir@saglik.gov.tr; 9Department of Gastroenterology, Ankara City Hospital, Ankara 06800, Türkiye; dilaraturan89@yahoo.com; 10Department of General Surgery, Ankara University, Ankara 06230, Türkiye; kirimker@ankara.edu.tr; 11Department of Nephrology, Ankara City Hospital, Ankara 06800, Türkiye; drezgi_76@hotmail.com; 12Department of Urology, Ankara City Hospital, Ankara 06800, Türkiye; erkanesin@mynet.com; 13Department of General Surgery, Transplantation, Faculty of Medicine, Başkent University, Ankara 06490, Türkiye; ebruayvazoglu@gmail.com (E.A.S.); rectorate@baskent.edu.tr (M.H.); 14Department of General Surgery, Adana City Hospital, Adana 01230, Türkiye; slmcetin@gmail.com; 15Department of Medical Microbiology, Koç University School of Medicine, Istanbul 34450, Türkiye

**Keywords:** solid organ transplantation, hematopoietic stem cell transplantation, mRNA vaccine, inactivated vaccine, neutralizing antibody, Anti-Spike IgG

## Abstract

We evaluated neutralizing antibodies against the Omicron variant and Anti-Spike IgG response in solid organ (SOT) or hematopoietic stem cell (HSTC) recipients after a third dose of BNT162b2 (BNT) or CoronaVac (CV) following two doses of CV. In total, 95 participants underwent SOT (*n* = 62; 44 liver, 18 kidney) or HSCT (*n* = 27; 5 allogeneic, 22 autologous) were included from five centers in Turkey. The median time between third doses and serum sampling was 154 days (range between 15 to 381). The vaccine-induced antibody responses of both neutralizing antibodies and Anti-Spike IgGs were assessed by plaque neutralizing assay and immunoassay, respectively. Neutralizing antibody and Anti-Spike IgG levels were significantly higher in transplant patients receiving BNT compared to those receiving CV (Geometric mean (GMT):26.76 vs. 10.89; *p* = 0.03 and 2116 Au/mL vs. 172.1 Au/mL; *p* < 0.001). Solid organ transplantation recipients, particularly liver transplant recipients, showed lower antibody levels than HSCT recipients. Thus, among HSCT recipients, the GMT after BNT was 91.29 and it was 15.81 in the SOT group (*p* < 0.001). In SOT, antibody levels after BNT in kidney transplantation recipients were significantly higher than those in liver transplantation recipients (GMT: 48.32 vs. 11.72) (*p* < 0.001). Moreover, the neutralizing antibody levels after CV were very low (GMT: 10.81) in kidney transplantation recipients and below the detection limit (<10) in liver transplant recipients. This study highlights the superiority of BNT responses against Omicron as a third dose among transplant recipients after two doses of CV. The lack of neutralizing antibodies against Omicron after CV in liver transplant recipients should be taken into consideration, particularly in countries where inactivated vaccines are available in addition to mRNA vaccines.

## 1. Introduction

Since the beginning of the pandemic, the novel coronavirus (SARS-CoV-2) has infected approximately 635 million individuals and caused almost 6.5 million deaths worldwide [1]. Hematopoietic stem cell transplantation (HSCT) and solid organ transplantation (SOT) recipients are at an increased risk for COVID-19 because of immunosuppressive medication and other comorbidities. Additionally, their responses to vaccination, which is the most effective way to control the pandemic, are substantially lower than those of immunocompetent individuals, occurring at rates comparable to those of unvaccinated individuals in some cases [2]. Despite the successful maintenance of vaccination processes, even past SARS-CoV-2 infection and post-vaccination immunity were not adequate to prevent infections with emerging variants, which highlights the significance of booster doses, particularly for high-risk individuals [3,4]. In 2022, a new variant of concern, Omicron, became dominant worldwide and raised concerns for the protective capacity of vaccination [5]. It also required a while to develop an immunization schedule for immunocompromised individuals, which is still being debated due to the needs and specific qualities of distinct immunocompromised states, such as HSCT and SOT.

SOT and HSCT recipients who had received two doses of inactivated vaccine, the only vaccine available in our country at the beginning of the pandemic, were given the option of receiving a third dose with either inactivated vaccine or mRNA vaccine. There has been limited research comparing the responses of mRNA vaccines and inactivated vaccines as a third dose in transplant recipients for the Omicron variant, and no studies comparing SOT and HSCT patients [6,7]. 

The aim of this study was to compare immunological responses to mRNA and inactivated vaccines against Omicron by measuring neutralizing antibodies and Anti-Spike IgG in SOT and HSCT recipients after the third dose, following two doses of inactivated vaccine administered 28 days apart. 

## 2. Materials and Methods

### 2.1. Study Design and Selection of Participants

In this multicenter observational study, 95 participants who underwent SOT (*n* = 44 liver transplantation, *n* = 18 kidney transplantation) or HSCT (*n* = 5 allogeneic HSCT, *n* = 22 autologous HSCT) with no history of COVID-19 were recruited. For patients with SOT, all living donors were blood relatives or spouses. Patients were selected from five centers in Turkey: Başkent Ankara Hospital, Ankara City Hospital, Ankara University School of Medicine, and Adana City Hospital. After collection, serum samples were transferred to the Koç University-İşbank Center for Infectious Diseases (KUISCID) for laboratory tests and stored at −80 °C until use. All participants included in the study received two doses of the inactivated vaccine, CoronaVac (CV), as their primary vaccination. All serum samples were collected 3–7 months after the third vaccine doses. Nineteen healthy control samples from volunteer Koç University Hospital healthcare workers (*n* = 10 for BNT third-dose receivers, *n* = 9 for CV third-dose receivers) were included in this study. Informed consent was obtained from all of the participants. This study was approved by the Başkent University Institutional Review Board (KA/22/84).

### 2.2. Cell Culture and Plaque Assay

Before testing, the serum samples were inactivated at 56 °C for 30 min. The SARS-CoV-2 Omicron variant (hcov-19/Turkey/koc_23122021_VK107/2021 (GISAID) Omicron BA.1.1), which was previously isolated from the SARS-CoV-2 RdRp PCR-positive nasopharyngeal specimen of a patient admitted to Koç University Hospital, was used for plaque neutralization assays.

Plaque neutralization assays were conducted under BSL-3 conditions. Vero E6 cells were cultured with DMEM High-Glucose (Sigma-Aldrich^®^, Burlington, MA, USA, cat. no: D6429) supplemented with 10% fetal bovine serum (FBS), (HyClone™ Logan, UT, USA, cat. no: SV30160.03HI), 1% Penicillin-Streptomycin (HyClone™ Logan, UT, USA, cat. no: SV30010), and Amphotericin B (HyClone™ Logan, UT, USA, cat. no: SV30078.01). Serial serum dilutions of 300 µL were incubated with 300 µL SARS-CoV-2 at a multiplicity of infection (MOI) 0.01 for 1 h at 37 °C, 5% CO_2_, and then 600 µL mixture was inoculated onto the Vero E6 cells at 100% confluency. After 1 h of incubation at 37 °C, 5% CO_2_, the serum-virus mixture was discarded. The cell monolayers were coated with 2% methylcellulose (Sigma-Aldrich^®^, Burlington, MA, USA, M0512, cat. no: 9004–67–5) and 5% FBS-DMEM mixture (1:1). Five days after infection, the methylcellulose/DMEM mixture was discarded. Plates were washed and cells were fixed with 4% PFA (Electron Microscopy Sciences, Philadelphia, PA, USA, cat. no: 15710-S) followed by Gram’s crystal violet solution staining (Merck Millipore, Darmstadt, Germany, cat. no: 109218). Plaques were counted with the naked eye, and the Celigo Image cytometer (Nexcelom, USA, Lawrence, Celigo Image Cytometer 200-BFFL-5C) and plaque reduction titers (PRNT50) were calculated. The viral control was studied in duplicate for each assay. 

### 2.3. Anti-SARS-CoV-2 Spike (S) IgG Measurement 

Anti-SARS-CoV-2 Spike (S) IgG was measured using the Abbott™ Alinity™ ci-series Integrated Clinical Chemistry and Immunoassay System (Abbott, Abbott Park, IL, USA, cat. no.04S1750) according to the manufacturer’s instructions. 

### 2.4. Statistical Analysis 

Statistical analysis of unpaired samples was performed using an unpaired nonparametric Mann–Whitney U test to compare two dependent groups. For the correlation analysis between neutralizing antibody and Anti-Spike IgG levels, the Pearson r correlation test was used. GraphPad Prism 8.0.2 Software was used for the analysis and visualization of the obtained data.

## 3. Results

### 3.1. Study Design and Overall Results

The overall study design is shown in Figure 1. Seventy (73.7%) of the 95 participants received BNT162b2 (BNT) as a booster dose (38 liver transplantation, 21 allogeneic or autologous HSCT, 11 kidney transplantation) and 25 (26.3%) of the participants received CV as a third dose (14 liver transplantation, 5 allogeneic and autologous HSCT, 6 from kidney transplantation). The median time between third doses and serum sampling was 154 days (IQR range between 15 to 381 days).

### 3.2. The Demographic Characteristics of the Participants

The demographic characteristics of the participants are presented in Table 1.

### 3.3. Overall Antibody Responses after the Third Dose in the Transplantation Group 

The neutralizing antibody levels against the Omicron variant in the transplantation group were significantly higher in BNT receivers than in CV receivers, with Geometric Mean Titer (GMT) levels of 26.76 for the BNT and 10.89 for the CV (*p* = 0.03). Likewise, the third dose of BNT showed significantly higher Anti-Spike IgG levels than CV with a GMT of 2116 Au/mL vs. 172.1 Au/mL, respectively (*p* < 0.001). In the healthy control group, BNT induced significantly higher neutralizing antibodies (GMT:50.40 vs. 10.48, *p* = 0.006) and Anti-Spike IgG levels (GMT: 29,328 AU/mL vs. 3204 AU/mL, *p* < 0.001) than CV (Figure 2). 

In the Pearson correlation test for neutralizing antibody and Anti-Spike antibody levels in the transplantation group, the correlation coefficient was found to be 0.95. Since the correlation coefficient falls between 0.8 and 1.0, the correlation was found to be very strong.

### 3.4. Antibody Responses after the Third Dose in the HSCT and the SOT Groups

In HSCT recipients, both vaccines elicited higher neutralizing antibody levels than SOT recipients. Thus, in the HSCT recipients, the GMT was 91.29 after BNT and it was 15.81 in the SOT group (*p* < 0.001). Likewise, the GMTs after CV were 34.82 and <10 in the HSTC and SOT recipients, respectively. In the HSCT recipients, BNT induced significantly higher levels of neutralizing antibodies than CV (GMT: 91.29 vs. 34.82) (*p* = 0.03). However, the difference in the Anti-Spike IgG levels after BNT and CV was not significant (GMT: 3259 vs. 720 AU/mL, *p* = 0.28) (Figure 3A,B). Among the SOT recipients, the antibody response after the third dose of CV was below the detection limit (GMT: <10). Furthermore, Anti-Spike IgG levels were found to be very low after CV (GMT: 120.3 AU/mL).

In the analysis of neutralizing antibody levels based on the duration after the third dose of vaccine, no significant decline was detected in either vaccine in either group. The neutralizing antibody response analysis showed that in the HSCT recipients with BNT, the GMT was reduced to 80 from 94 after 6 months. In the SOT recipients with BNT, the GMTs were 15.80 and 20 before and after 6 months (*p* = 0.7271). The Anti-Spike IgG levels were at the same trend with those of the neutralizing antibodies. In the SOT group, BNT elucidated significantly lower neutralizing antibody levels (*p* = 0.0014) and IgG levels (*p* = 0.0004). Similarly, the IgG response with CoronaVac in the control group was found to be significantly higher than in the SOT group (*p* = 0.002).

### 3.5. Antibody Responses after the Third Dose in Each Transplantation Group 

Lastly, we evaluated the neutralizing antibody and Anti Spike IgG levels based on transplantation types (liver, kidney, allogeneic, and autologous HSCT) (Figure 4A,B). 

In the auto-SCT recipients, BNT induced a significantly higher neutralizing antibody response (GMT: 108.3) than CV (34.82) (*p* = 0.04), while the increase in the Anti-Spike IgG levels was not significant (GMT: 1879 Au/mL vs. 720.8 Au/mL, respectively, *p* = 0.55).

The neutralizing antibody responses to BNT in kidney transplantation recipients were significantly higher than those in liver transplantation recipients (GMT:48.32 vs. 11.72) (*p* < 0.001). The CV elicited a weak neutralizing antibody production in kidney transplantation recipients (GMT: 10.81) and the antibody levels were below the detection limit (<10) in liver transplant recipients. Likewise, Anti-Spike IgG levels in liver transplant recipients were found to be very low (GMT: 97.04 AU/mL) after CV. 

In the comparison of vaccine efficacy in kidney transplantation recipients, BNT induced significantly higher neutralizing and Anti-Spike IgG antibody (GMT: 48.32 and GMT: 1962 AU/mL) than CV (GMT: 10.81 and GMT: 198.6 AU/mL) (*p* = 0.01 and *p* = 0.03). 

Immunosuppressive regimens in SOT recipients revealed no significant differences in neutralizing activity and Anti-Spike IgG levels between the antimetabolite-using and non-using groups (*p* = 0.061 and 0.682, respectively). Among the HSCT recipients, only three patients received immunosuppressive therapy.

## 4. Discussion

In this multicenter study, including SOT and HSCT recipients who received two doses of CV administered 28 days apart, the GMT of neutralizing antibody against omicron was found to be 2.45-fold higher and the GMT of anti-spike IgG antibody was found to be 18-fold higher after the third dose of BNT than after the third dose of CV. Compared with CV, BNT had considerably better responses in both groups of transplant patients. Solid organ transplantation patients, particularly liver transplant recipients, showed lower antibody levels than HSCT recipients. A lower humoral response was not found to be related to the immunosuppressive regimen, contrary to current publications [8,9,10,11]. For BNT recipients of HSCT, all individuals had positive neutralizing antibody titers regardless of the time elapsed after vaccination. In the autologous HSCT group, the immune responses were similar to those in the healthy controls. Despite the fact that BNT positivity remained consistent in SOT recipients before and after 6 months, no positivity of neutralizing antibodies was seen in any period with CV. While the neutralizing antibodies were negative, the anti-spike IgG antibodies were positive. 

In phase 2/3 research and field tests conducted at the beginning of the outbreak in healthy adults, inactivated vaccines were found to be effective in preventing symptomatic or severe illness [12,13,14]. However, as research on immunosuppressive patients grew and novel variants appeared, several studies have demonstrated that mRNA vaccines are more protective against illness and more effective for hospitalization and death in both immunosuppressive patients and the general population [2,15,16]. After recognizing that antibody responses to SARS-CoV-2 decrease over time, additional doses are needed in both healthy and immunosuppressed individuals. Crucially, in immunocompromised patient groups, such as SOT or HSCT, antibody responses have been shown to decline substantially sooner than in healthy controls [17]. Based on this finding, the CDC recommended additional doses and defined the primary scheme for this patient group as three doses, beginning in October 2021 [18]. In addition, vaccine effectiveness was found to be better with three doses than two doses of SARS-CoV-2 vaccines according to hospitalization, intensive care requirement, and mortality [19].

Recent studies indicated that heterologous vaccination (BNT after two doses of CV) provides neutralizing activity higher than three doses of CV in healthy adults [20]. These findings are similar to the responses developed after two doses of BNT and protective levels for new emerging variants, such as Omicron, which can escape from the vaccine or infection-induced immunity [21]. However, there are limited studies on transplant recipients that show differences in immunological responses in countries where various vaccine types are available. Our study was different in that the testing for neutralizing antibodies was against the current variant Omicron. Dib et al. reported a better humoral response after three doses of mRNA vaccines than after heterologous regimens in SOT recipients [22]. However, in countries in which only inactivated vaccines were available at the beginning of the pandemic, patients had access to an mRNA vaccine only as the third dose, and studies indicated better immunogenicity after mRNA boosters. Pestana et al. reported higher seroprevalence, seroconversion, and IgG antibody values with a booster with BNT after two doses of CV as a primary regimen in kidney transplant recipients [8]. In our study, similar to other recent studies, additional BNT was found to be more effective than CV after two doses of CV were administered 28 days apart regarding the type of transplantation and time passed after vaccination [21]. This finding is particularly notable in countries where there is an availability of a variety of vaccinations with varying efficacy in preventing illness onset, hospitalization, and mortality.

According to current studies, immune responses among HSCT recipients after COVID-19 vaccination are better than immune responses among SOT recipients, and the durations of protection in autologous transplant recipients are higher than those in allogeneic transplant recipients, in line with our study [9,19,23,24,25]. These results are important because current vaccination schemes are prepared without considering the reasons for immunosuppression and differences in vaccine type. 

Previous studies have shown a strong correlation between neutralizing activity and Anti-Spike IgG levels [13,26]. Similarly, our results show a strong correlation (correlation coefficient: 0.94) between the neutralizing activity and Anti-Spike IgG levels. 

The strengths of our study are the comparison of immune responses following different COVID-19 vaccines in various transplant types, the evaluation of spike antibody and neutralizing antibody activities against mRNA and inactivated vaccines, and the evaluation of neutralizing antibody responses against the current variant Omicron.

The limitations of our study are the small number of participants and the absence of real-life data. Additionally, we were unable to analyze T-cell responses during the study period. At the time of the study, there were no established recommendations for COVID-19 vaccination for the immunosuppressed population; therefore, we could not define additional doses as boosters or the last dose of the primary scheme. 

## 5. Conclusions

This study highlights the superiority of BNT responses as the third dose when compared with CV responses among SOT and HSCT recipients after two doses of CV. Emerging variants are of pivotal importance for the protection levels of vaccines in real life; therefore, the effects of the variants should be taken into consideration in neutralization studies. These findings are more significant in countries such as Turkey, where inactivated vaccines are available, in addition to mRNA vaccines. Further studies are needed to establish vaccination schedules for immunosuppressed groups in various countries.

## Figures and Tables

**Figure 1 viruses-15-01534-f001:**
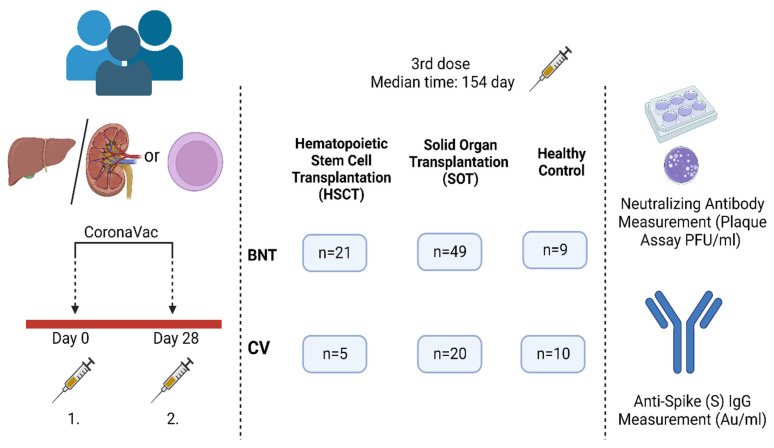
The participants and experimental workflow of the study.

**Figure 2 viruses-15-01534-f002:**
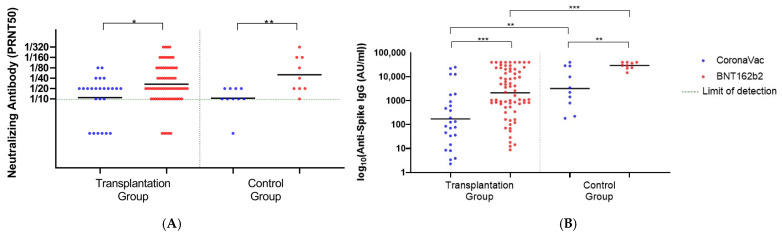
The neutralizing and Anti Spike IgG levels after the third dose of BNT162b2 or CoronaVac. (**A**) The neutralizing antibody (PRNT50) titers of transplantation (*n* = 25 for CoronaVac; 70 for BNT162b2) and health control groups (*n* = 10 for CoronaVac; 9 for BNT162b2). (**B**) The Anti Spike IgG levels of transplantation and health control groups. (* indicates *p* < 0.05, ** indicates *p* < 0.01, *** indicates *p* < 0.001). Black lines indicate the GMTs and each dot represents a single individual.

**Figure 3 viruses-15-01534-f003:**
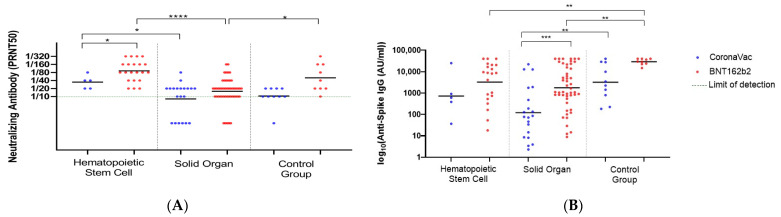
The neutralizing and anti-Spike IgG levels after the third dose of vaccination in HSCT and SOT. (**A**) The neutralizing antibody (PRNT50) titers of the HSCT (*n* = 5 for CoronaVac; 21 for BNT162b2) and SOT group. (*n* = 20 for CoronaVac; 49 for BNT162b2). (**B**) The Anti Spike IgG levels of the HSCT and SOT group (* indicates *p* < 0.05, ** indicates *p* < 0.01, *** indicates *p* < 0.001, **** indicates *p* < 0.0001). Black lines indicate the GMTs and each dot represents a single individual.

**Figure 4 viruses-15-01534-f004:**
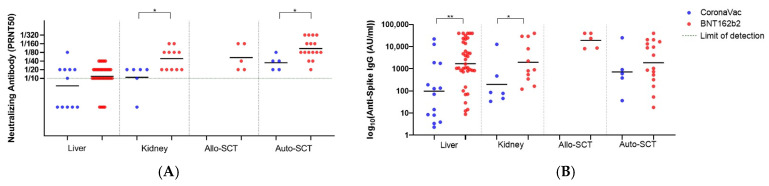
The neutralizing and Anti Spike IgG levels after the third dose of vaccination for each transplantation group. (**A**) The neutralizing levels after the third dose of vaccination for each transplantation group and healthy control group. (**B**) Anti Spike IgG levels after the third dose of vaccination for each transplantation group and healthy control group (* indicates *p* < 0.05, ** indicates *p* < 0.01). Black lines indicate the GMTs and each dot represents a single individual. (Liver group *n* = 14 for CoronaVac; 38 for BNT162b2), (Kidney group *n* = 6 for CoronaVac; 11 for BNT162b2), (Allo-SCT *n* = 5 for BNT162b2), (Auto-SCT *n* = 5 for CoronaVac; 16 for BNT162b2).

**Table 1 viruses-15-01534-t001:** The demographic characteristics of the study group.

	Total (*n* = 95)	BNT162b2 Group (*n* = 70)	CoronaVac Group (*n* = 25)
Median age (IQR *) Age group (*n*/%) 18–29 30–39 40–49 50–59 >60	56 (42–63) 9 (9.5) 12 (12.6) 18 (18.9) 24 (25.3) 32 (33.7)	56.5 (43–65) 6 (8.6) 9 (12.9) 13 (18.6) 16 (22.9) 26 (37.1)	52 (39.5–60) 3 (12.0) 3 (12.0) 5 (20.0) 8 (32.0) 6 (24.0)
Female gender (*n*/%)	28 (29.5)	24 (34.3)	4 (16.0)
Type of transplantation (*n*/%) SOT ** Liver Kidney HSCT *** Autologous Allogeneic	69 (72.6) 52 (54.7) 17 (17.9) 26 (27.4) 21 (22.1) 5 (5.3)	49 (70.0) 38 (54.3) 11 (15.7) 21 (30.0) 16 (22.9) 5 (7.1)	20 (80.0) 14 (56.0) 6 (24.0) 5 (20.0) 5 (20.0) 0 (0)
Median time after transplantation-years (IQR)	4 (2–6)	4 (2–6)	4 (3–6)
Time after the booster dose (*n*/%) <6 months ≥6 months	82 (86.3) 13 (13.7)	60 (85.7) 10 (14.3)	22 (88.0) 3 (12.0)
Antimetabolite usage for SOT (*n*/%)	28 (40.6)	23 (32.9)	5 (20.0)

* Interquartile range; IQR; ** Solid organ transplantation; SOT; *** Hematopoietic stem cell transplantation; HSCT.

## Data Availability

The corresponding authors (Füsun Can and Özlem Kurt Azap), upon request, will provide information supporting the conclusions of this study. Owing to patient privacy, the data are not publicly accessible.

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
