# Peer review of "BNT162b2 or CoronaVac as the Third Dose against Omicron: Neutralizing Antibody Responses among Transplant Recipients Who Had Received Two Doses of CoronaVac"

_viruses, 2023, doi:10.3390/v15071534_

Round 1

Reviewer 1 Report

This manuscript investigated BNT162b2 or CoronaVac as the Third Dose against Omicron:

Neutralizing Antibody Responses Among Transplant Recipients Who Had Received Two Doses of CoronaVac. The  effectiveness of mRNA vaccine and inactivated vaccine on different groups especially for the immunosuppressed population is a very interesting topic. But during SAS-Cov-2 variants quickly iteration, new vaccines such as both Pfizer-BioNTech and Moderna COVID-19 bivalent mRNA vaccines were authorized. Thus application significance of this study was decreasing.

There are some other issues in this manuscript.

1. The control was missing before the third dose.

2. The authors ignored natural infection in realization world.

3. The data of control group should be included in both figure 3 and 4 and difference and further discussed.

4. The different humoral immunity in transplantation group should be further analyzed.

 Minor editing of English language required

Author Response

Dear Editor,

We would like to thank you for your response and comments on our manuscript titled ‘BNT162b2 or CoronaVac as the Third Dose against Omicron: Neutralizing Antibody Responses Among Transplant Recipients Who Had Received Two Doses of CoronaVac’

We adressed all comments of the rewiewer’s and presented our point by point responses below:

Reviewer 1:

  1. Comment: The control was missing before the third dose.

Response: We included the groups only after third dose of vaccines.  There is no analysis before 3rd dose in the study group as well.

  1. Comment: the authors ignored natural infection in realization world.

Response: The natural infection history was questioned as presented in the methods section “In this multicenter observational study, 95 participants who underwent SOT or HSCT with no history of COVID-19 were recruited”

  1. Comment: The data of control group should be included in both figure 3 and 4 and difference and further discussed.

Response: The control group was added in to Figure 3 and discussed accordingly.

  1. Comment: The different humoral immunity in transplantation group should be further analyzed.

Response: The humoral immunity was analyzed regarding Anti-spike and neutralizing antibody responses in transplantation group in different transplantation types as shown in figure 4. If the reviewer suggests B cell response, unfortunately we did not have whole blood for flow.

Reviewer 2 Report

This study aims to compare the immunological responses of solid organ transplantation (SOT) and hematopoietic stem cell transplantation (HSCT) recipients to mRNA and inactivated vaccines against the Omicron variant of the novel coronavirus. SOT and HSCT recipients, who are at increased risk for COVID-19 due to immunosuppression, received two doses of the inactivated vaccine and were given the option of a third dose with either inactivated or mRNA vaccine. The study measured neutralizing antibodies and Anti-Spike IgG levels in these recipients to evaluate their immune responses after the third dose. The research addresses the need to understand vaccine responses in immunocompromised individuals and provides insights into the effectiveness of different vaccine types in this population.

Minor question:

1.      Please pay attention to the writing format of punctuation, like line84. Please use a complete sentence to explain the sample number or include the parentheses in the first sentence.

2.      Please add sample size to the legend of Figure1,2,3,4.

1.      Please consider adding the age distribution in Table 1.

Author Response

Dear Editor,

We would like to thank you for your response and comments on our manuscript titled ‘BNT162b2 or CoronaVac as the Third Dose against Omicron: Neutralizing Antibody Responses Among Transplant Recipients Who Had Received Two Doses of CoronaVac’

We adressed all comments of the rewiewer’s and presented our point by point responses below:

Reviewer 2:

  1. Comment: Please pay attention to the writing format of punctuation, like line84. Please use a complete sentence to explain the sample number or include the parentheses in the first sentence.

Response: Corrected accordingly.

  1. Comment: Please add sample size to the legend of Figure 1,2,3,4.

Response:  Corrected as recommended.

  1. Comment: Please consider adding the age distribution in Table 1.

Response:  Corrected as recommended.

Round 2

Reviewer 1 Report

Although the author added the control group, but the results showed there are not good correlation between spike IgG and neutralization antibody.  

Quality of English Language is good.

Author Response

Dear Editor,

We would like to thank you for your response and comments on our manuscript titled ‘BNT162b2 or CoronaVac as the Third Dose against Omicron: Neutralizing Antibody Responses Among Transplant Recipients Who Had Received Two Doses of CoronaVac’

We adressed all comments of the rewiewer’s and presented our point by point responses below:

Reviewer 1:

  1. Comment: Although the author added the control group, but the results showed there are not good correlation between spike IgG and neutralization antibody.

Response: The correlation coefficient is found to be 0.94. In the results section between the line 167-160 the strong correlation was described. In the discussion section the interpretation was corrected.